# Data-Driven Fault Detection and Diagnosis: Challenges and Opportunities in Real-World Scenarios

Francesca Calabrese *, Alberto Regattieri ⬤, Marco Bortolini ⬤ and Francesco Gabriele Galizia ⬤

Department of Industrial Engineering (DIN), University of Bologna, 40136 Bologna, Italy
* Correspondence: francesca.calabrese9@unibo.it

**Abstract:** The pervasive digital innovation of the last decades has led to a remarkable transformation of maintenance strategies. The data collected from machinery and the extraction of valuable information through machine learning (ML) have assumed a crucial role. As a result, data-driven predictive maintenance (PdM) has received significant attention from academics and industries. However, practical issues are limiting the implementation of PdM in manufacturing plants. These issues are related to the availability, quantity, and completeness of the collected data, which do not contain all machinery health conditions, are often unprovided with the contextual information needed by ML models, and are huge in terms of gigabytes per minute. As an extension of previous work by the authors, this paper aims to validate the methodology for streaming fault and novelty detection that reduces the quantity of data to transfer and store, allows the automatic collection of contextual information, and recognizes novel system behaviors. Five distinct datasets are collected from the field, and results show that streaming and incremental clustering-based approaches are effective tools for obtaining labeled datasets and real-time feedback on the machinery's health condition.

**Keywords:** predictive maintenance; industrial application; fault detection and diagnosis; novelty detection

## 1. Introduction

Maintenance assumes a critical role in manufacturing companies' productivity and profitability [1,2] and has a significant impact on their economic, environmental, and social dimensions [3]. In 2016, maintenance expenditures and avoidable losses in discrete manufacturing were estimated at USD 193.6 billion [4]. In addition, lousy maintenance may compromise the quality of final products and the safety of working environments. For these reasons, maintenance strategies have evolved from simply repairing an asset after a failure to predicting the time when equipment will fail to intervene before its breakdown.

Predictive maintenance (PdM) is determining maintenance actions according to regular inspections of equipment's physical parameters and degradation mechanisms to intervene on the machine before its failure [5]. Relevant parameters, such as vibrations, acoustic emissions, currents, pressures, temperatures, and others, are monitored, and the health condition of the system is assessed at any point in time based on their historical and current values. In addition, its future health conditions are predicted to compute its remaining useful life (RUL) and schedule maintenance interventions accordingly. This process is often referred to as prognostics and health management (PHM) in the literature and consists of four main steps: data collection, feature extraction, diagnostics, and prognostics [6].

In the last decades, scientific and industrial interests in PHM have grown, together with information and communication technologies (ICT) and digital innovation of factories, whose elements fall into the concept of Industry 4.0. In this context, machines are connected to the Industrial Internet of Things (IIoT) that enables the collection of massive data from physical assets on the shop floor, their transmission into the cloud, and their processing with intelligent algorithms to make factories more productive, less costly to operate, and

more reliable [7]. On the one hand, knowing the system's health condition in a given instant leads to maximizing the production profits and minimizing all costs and losses by implementing efficient and just-in-time maintenance strategies [8]. On the other hand, data-driven PdM requires the collection of large volumes of historical data from multiple sources and proper big data processing technologies to build an integrated environment in which production and maintenance processes can be managed more efficiently [9].

The large body of existing literature on PdM mainly focuses on improving the accuracy of data processing through machine learning (ML) and deep learning (DL) algorithms. In particular, several data-driven fault detection and diagnosis approaches are developed and applied to different domains. Supervised models, such as shallow or deep artificial neural networks (ANNs) [10,11], are largely adopted for rotating machinery fault diagnosis. In these applications, signals are collected from the component in one or more health conditions and the goal is to build a classification model that classifies future signals into one of those conditions with the highest accuracy. Therefore, all fault classes are known in advance. Unsupervised models, such as k-means [12] or autoencoders [13], are used for fault detection, whose aim is to automatically detect any fault in its early stage [14]. In these applications, only two classes, health or damaged, are often considered. Therefore, diagnosing the detected fault is not automatic but requires human intervention. In addition, industrial artificial intelligence deals with industrial big data, characterized by the so-called 3Bs challenges, broken data, bad quality data, and background [15]. In the big challenge to fit the AI in industries and implement data-driven PdM, the more significant issues are related to the large amount of data generated by machines, the lack of contextual information, the unbalanced datasets, and the dynamic nature of industrial environments [16]. Consequently, latency, bandwidth, and network scalability limit the real-time monitoring of machinery. The absence of labels indicating the actual health condition and the lack of data in all faulty modes restrict the use of supervised ML models [17]. Finally, the accuracy of these models reduces with time because of their poor ability to learn from current data and adapt to multiple scenarios. Given these issues, the traditional offline and batch fault diagnosis, which relies on many labeled data, may be ineffective and impracticable in real-world applications. In other words, the system health management has to handle unlabeled and unbalanced datasets and operate in an evolving environment. These two aspects are strictly correlated and can be addressed by integrating offline and batch processing with online and streaming models to recognize both known conditions (fault diagnosis) and detect unknown behaviors (novelty detection) [18]. Offline batch processing aims to uncover valuable information from accumulated data for fault prediction and is suitable for stored data, as it mainly focuses on the accuracy and comprehensiveness of analysis instead of focusing on real-time fault detection and diagnosis. On the other hand, real-time streaming processing is needed in variable and complex operating conditions to realize a real-time diagnosis and ensure timely maintenance. In a previous work [19], the authors proposed a novel framework for implementing PdM. In particular, the framework consists of an edge layer, in which raw signals are collected and processed to extract relevant information at a component and system level (i.e., feature extraction); in addition, novelty detection algorithms are also applied to streaming data to discover new system behaviors and component faults. This information is finally collected in the cloud, where supervised ML models are built on historical data and updated based on newly collected data.

This paper is an extension of [19], in which the proposed framework is validated on a dataset collected from a test rig built in the Department of Industrial Engineering of the University of Bologna, Italy, and on four real industrial datasets, which differ for the components under analysis, the kind of collected signals, the quantity of the data, and the number of health and fault conditions.

The remainder of the paper is organized as follows. Section 2 describes the research questions and the main contributions of this paper. Section 3 provides an overview of the existing literature on fault detection and diagnosis using incremental learning for streaming data. Section 4 describes the available datasets and the experimental scenario. Section 5

shows the results obtained by applying a clustering-based approach for novelty detection. Finally, Section 6 concludes the work by discussing the implications from the industrial point of view and providing future research directions.

## 2. Research Questions and Contributions

The core of a data-driven PdM strategy is transforming raw data into useful information to support the decision-making process. Although ML and DL methods are powerful, industrial contexts impose some constraints on their use in the PdM field for several reasons. The first limitation is the lack of a systematic methodology for collecting and managing the massive data available from one or more plants. The second limitation is the lack of labeled historical data and industrial environments' dynamic and heterogeneous nature.

Concerning the first point, the main issues concern the types of data that need to be available for building accurate and robust models for fault detection and diagnosis, the time at which the data are required, and how they can be organized for facilitating the preprocessing phase. Indeed, the data are often physically taken from the machinery and stored in a PC, in which ML models for fault diagnosis are built. However, these datasets are unlabeled, i.e., the health condition associated with the condition monitoring data and the system working condition, and are not included. Therefore, a time-consuming phase is necessary for assigning the correct label to all observations. In addition, the massive amount of condition monitoring data leads to collecting only a few minutes of data per day, limiting the use of complex classification models and the analysis of the system behavior.

Concerning the second point, the main issues concern data availability and the ability to learn new behaviors as they occur. Indeed, during the training phase, classification models for fault diagnosis learn the relationships between the condition monitoring data and the corresponding health condition. Therefore, when applied to new data, they can only classify the observations into the classes involved in the learning process. However, not all fault conditions are known in advance or can be simulated because of safety issues. Hence, these models fail to recognize and classify observations corresponding to a health condition not included in the training set. In addition, industrial contexts are inherently dynamic, which makes classification models trained on historical data obsolete and less accurate as the operating condition of a system changes or a modification of the context of use is introduced.

Given these issues, the research questions (RQs) that this paper wants to address can be summarized as follows:

1. RQ1. Which data are needed for building an accurate and robust predictive model for diagnostics? How should the dataset be organized?
2. RQ2. How can labels be automatically obtained, considering that novel behaviors can occur during machine functioning?

Underpinned by RQ1, the first purpose of this paper is to exploit the experience derived from several collaborations with industries in the Emilia Romagna region (Italy) and the knowledge acquired from the existing literature for proposing a classification of the available data and explaining how each kind of extracted data can contribute to building a health management system. RQ2 requires acting at the data analysis level, focusing on different approaches and learning paradigms to integrate historical information with streaming data to provide real-time feedback on the machinery's health condition for the maintenance actions scheduling. Hence, the second purpose of this paper is to validate a general method for the automatic labeling of streaming data that can reduce the data preprocessing and facilitate the data analysis that creates value and supports the decision-making process. Finally, a transversal goal of this paper is to provide knowledge to scholars and practitioners on the criticalities affecting the realization of a PdM strategy for the health management of industrial assets.

The main contributions of the present work may be summarized as follows:

1. The definition of the data and information flows in an edge-cloud-based PdM system;

2. The validation of novelty detection approaches on five distinct datasets from different industrial contexts, with a particular focus on the tasks performed on streaming data leading to the reduction in the quantity of data and the automatic labeling of collected datasets.

## 3. Related Works

### 3.1. Data Collection

According to [20], the types of data used for predictive maintenance are maintenance history (73%), asset usage (72%), asset condition (71%), condition data, and maintenance history of other assets within the company (42%), environmental data (29%), condition data, and maintenance history of other assets from other companies (9%), and others (7%). Tiddens et al. [21] grouped the data required for predictive maintenance into four categories: (1) asset history data, which include historical records of failures, inspections, and the related costs; (2) usage and process data, which entail operational data, e.g., running hours, mileage, or tons produced; (3) stressor data, which include the operating conditions of the system, such as the load exerted on the system, and environmental data; and (4) condition monitoring data, which include signals able to reveal the system's health conditions and provide indication of failures. Although all these data may contribute to implementing a predictive maintenance strategy, intelligent fault diagnosis only relies on condition monitoring data in practice. In particular, vibration signals have been used most frequently for machinery health management, followed by acoustic emissions, force/torque, temperature, and electric signals [22]. In a few cases, the data are directly provided by machine log systems, without installing sensors for the collection of condition monitoring data [23]. However, the number of studies exploring other types of data for predictive maintenance and the industrial applications that exploit the Industry 4.0 technologies for intelligent maintenance is limited.

Condition monitoring data, such as force, vibration, temperature, voltage, and others, reveal the health state of equipment working under certain operating condition, and are fundamental to building a fault diagnosis model, estimating its parameters, and verifying/validating its maturity [24]. However, raw data are redundant and noisy and cannot be directly used for fault diagnosis. Therefore, when using a data-driven approach for diagnostics and prognostics, processing and transforming raw signals is a fundamental step [25]. In particular, it is crucial to extract some characteristics from raw signals, named features, which must be relevant and nonredundant. The relevance of a feature concerns its ability to distinguish the different healthy and faulty conditions of the system under analysis. The set of features must contain the minimum informative content, i.e., it has to be nonredundant, to reduce the computational complexity of machine learning (ML) algorithms used for fault diagnosis.

Feature extraction and dimensionality reduction notably reduce the amount of data, which may have an important implication from the data sharing and storage perspective [26,27]. Through the IIoT architecture, it is possible to collect the data from all machines on a shop floor and query streaming data used for machine learning models [28]. They can be trained by resorting to the cloud computing paradigm, which represents a hosting platform that provides diagnosis solutions as a service [29]. However, the transmission of massive data to the cloud can generate high response times, which is unsuitable for fault detection systems for which real-time responses are essential. Performing data processing at the edge may solve this issue [30] because only features extracted at the edge may be sent to the cloud [31]. In addition, the reduced amount of data from the transmission into the cloud of relevant features may allow a more accessible aggregation of data collected from similar machines installed in different plants, obtaining larger datasets for model training. Therefore, a proper infrastructure should include sensor networks, edge devices, and cloud data centers to distribute the computing among different "devices" and reduce the latency of both transferring and inference and the required storage memory.

Several IoT architectures have been proposed in the literature for data collection and PdM. However, to the best of our knowledge, none of them explains the data and information to collect and how.

### 3.2. Streaming Novelty Detection

According to [32], two strategies exist to cope with unlabeled and unbalanced datasets. The first one considers evolving algorithms, according to which the learning phase is repeated as required to manage state detection (as new data are available), especially in case of unbalanced data. The second strategy deals with incompleteness and uncertainty of labels through semi-supervised learning approaches. A solution that integrates the two strategies may consider novelty detection in the evolving learning framework [33].

In evolving environments, observations arrive in a data stream, with no labels except for the initial training set [34]. When unlabeled observations arrive in the data streams, new labels may emerge. The main goal of incremental learning is to recognize abnormal conditions (anomalies) that have never occurred before while classifying machinery conditions (nominal and fault health conditions) as one of the known classes if there is no novel abnormality [35]. In other words, incremental learning approaches allow detection of the so-called concept drift, which occurs in the data stream when a machinery condition changes. In addition, they should also determine whether the concept drift corresponds to a known machinery operating condition or a novel operating condition. This problem, known as novelty detection in a data stream, is rarely addressed in the literature [36]. In other words, novelty detection deals with detecting new data samples that an ML model was not previously aware of [37,38].

Novelty detection is traditionally seen as a one-class classification problem in which only the nominal class is available. Unlike anomaly detection, novelty detection does not look for one single observation differing from the nominal data. On the contrary, it looks for a set of points that are not explained by the diagnosis model [39]. In addition, novelty detection recognizes novel behaviors when two or more classes exist. In these cases, a condition is considered novel when it differs from all known classes [40]. In these cases, the problem is recognizing novelties and classifying the known observations into one of the existing classes. In [41], a threshold-based classifier is adopted, which defines a fault as unknown when none of the classifier output exceeds a defined threshold. Cariño, J.A. et al. [42] introduce a methodology to discover new patterns when only nominal data are available. The methodology consists of an ensemble-based classifier, used for novelty detection, and an evolving classifier that is used for diagnosis. Similarly, the hybrid approach introduced in [43] aims at diagnosing known faults and detecting unknown faults using time–frequency features. In [44], the authors introduce a novel algorithm that learns drifting concepts from streaming and unlabeled data. An application of that algorithm to fault diagnosis is proposed in [45], where gradual and abrupt changes are first detected, and a classifier is then updated to include the new class.

Other approaches to novelty detection are based on semi-supervised and unsupervised learning. In the first category, several clustering algorithms for streaming processing have been proposed, such as OnLine Novelty Detection and Drift Detection Algorithm (OLINDDA) [46], Higia [47], and MultIclass learNing Algorithm for data Streams (MINAS) [48]. Basically, these algorithms generate a model using labeled data during an offline training phase; then, novel patterns made of examples that the model does not explain are recognized as unknown sets during the online phase. The second category of clustering-based approaches for novelty detection does not require any training phase. These models rely on the idea that distinct clusters correspond to distinct system conditions [49]. Therefore, each point can be either integrated into one existing clustering if the data are compatible with the existing model structure or generate a new cluster. In the first case, the parameters of existing clusters are updated to include the information provided by the last point [50]. In [51], a recursive algorithm for clustering and drift detection is proposed, in which clusters are updated according to the similarity between the data

and the changes occurring in the data flow. For this reason, the algorithm is more robust to outliers and noise. However, these algorithms need to know the cluster membership function in advance. In contrast, in [52], the membership function based on the input data density and distribution, therefore, is extracted from the data, and represents their density and distribution. In this context, the concept of cluster is replaced by the concept of cloud, which, unlike clusters, has no boundaries, could have any shape, and does not have a center. Therefore, the variables affecting the cloud shape and the assignment of points to the cloud are the local and global density, which are calculated recursively as new data arrive. Based on these concepts, ref. [53] proposed a two-step methodology for fault detection and identification in streaming data. First, faults are detected through autonomous anomalous detection (AAD). Then, a fuzzy classifier named AutoClass assigns the data sample to existing clouds or new clouds, depending on their closeness to the cloud center. Contrary to the AutoClass algorithm, the autonomous data partitioning (ADP) algorithm proposed in [54] is an evolving clustering algorithm that does not impose a data generation model on the empirical observation and is free from user-specific parameters.

To summarize, novelty detection approaches may be classified according to the learning paradigm they adopt, which can be supervised or unsupervised, and the necessity of a training phase before the streaming application. Consequently, three approaches can be distinguished:

1.  Classification-based approaches that require a training phase on one or more classes (e.g., COMPOSE);
2.  Clustering-based approaches that require a training phase on one or more classes (e.g., OLINDDA and MINAS);
3.  Clustering-based approaches that can be applied from scratch (e.g., ADP).

More recently, deep learning models, such as autoencoders and long short-term memory (LSTM) networks [55,56], are used to detect novelties. However, they require a large amount of data and a high computational effort for training. In addition, they have to be retrained each time a novel behavior is detected [57].

### 4. Experimental Design

In this section, five different datasets are analyzed. They have been collected from different automatic machinery during tests simulating nominal and improper working conditions of one or more components. The components under analysis, the collected signals, the health conditions, the corresponding durations, and the sampling frequency of raw signals for each dataset are summarized in Table 1.

**Table 1.** Description of the available datasets.

| Dataset | Environment | Component/s | Signals | Health Conditions | Dataset Duration | Raw Data Sampling Frequency (Hz) |
|---|---|---|---|---|---|---|
| Dataset 1 | Laboratory | Electric Motor | Vibration | Nominal 1<br>Nominal 2<br>Nominal 3<br>Fault 1 | 82.8 min<br>184.2 min<br>77.4 min<br>25.8 min | 12,800 |
| Dataset 2 | Industry | Case packer | Current | Nominal<br>Fault 1<br>Fault 2<br>Fault 3<br>Fault 4 | 15 s<br>15 s<br>15 s<br>15 s<br>7.05 s | 500 |
| Dataset 3 | Industry | Suction cups | Pressure | Nominal<br>Fault s1<br>Fault s2 | 15.99 s<br>11.65 s<br>3.52 s | 10,000 |
| Dataset 4 | Industry | Extruder | Temperature | Nominal 1<br>Nominal 2 | 67.5 (M1)–8196.5 (M2) h<br>796 (M1)–326 (M2) h | 1 |
| Dataset 5 | Industry | Sealing Group | Displacement | Nominal<br>Fault s1<br>Fault s2 | 266.4 s<br>160 s<br>152 s | 1000 |

### 4.1. Dataset 1: Electric Motor

The first dataset was collected from a test rig built in the laboratory of the Department of Industrial Engineering, University of Bologna, Italy. The platform includes an asynchronous motor, a gearbox consisting of two pulleys that exchange the rotation through a belt, two shafts that share the motion thanks to a couple of gears, and an electromagnetic brake. The full description of the system can be found in [19]. Three triaxial accelerometers, characterized by an acceleration range of 500 Gpeak, are placed on the system. The first one is placed on the bearing's support; the second one is placed next to the second pulley; and the third one is placed close to the gearboxes. Data are collected with a sampling frequency of 12.8 kHz.

The available dataset includes vibration signals collected during tests simulating four distinct operating conditions: Nominal 1, Nominal 2, Nominal 3, and Fault 1. In the first three conditions, all components of the test rig, i.e., the electric motor, the belt, the pulleys, and the gears, are in optimal conditions. Different values of the distance between the pulleys and the braking torque determine the implemented operating condition. During the fourth test, a sudden failure occurred to the electric motor. The first 5 min of data collected in each condition are depicted in Figure 1.

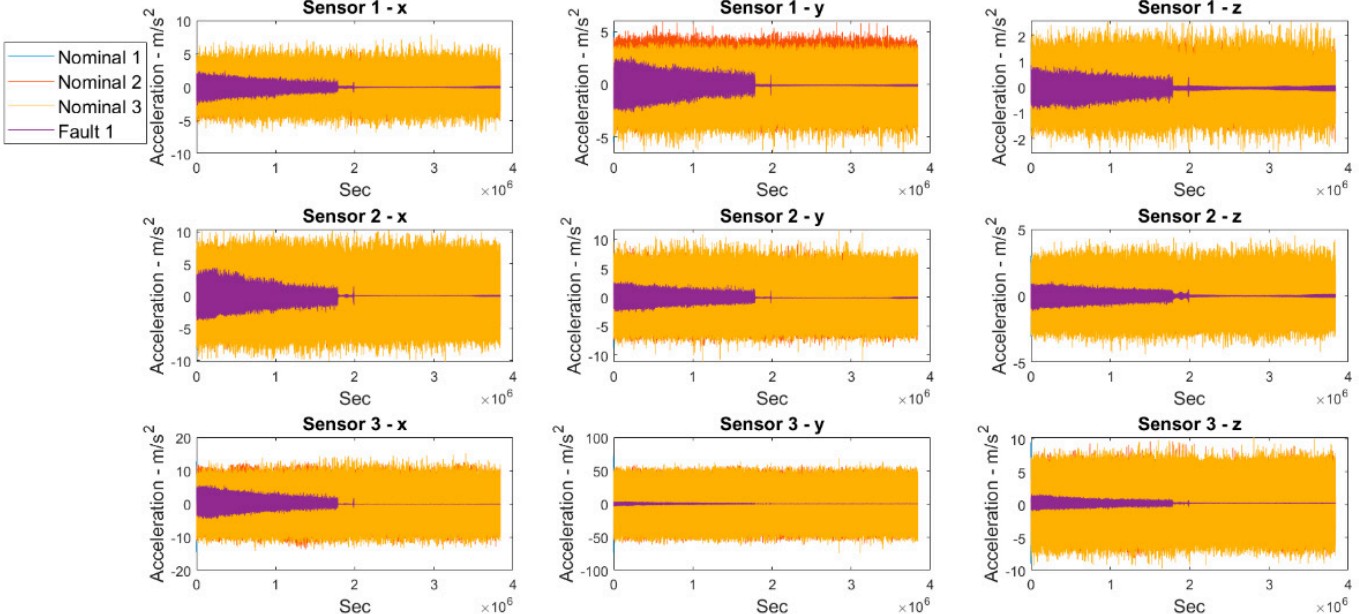

**Figure 1.** Raw signals generated by the system during the three nominal conditions and the fault condition.

### 4.2. Dataset 2: Case Packer

In this case, a subgroup of an automatic industrial machine is considered. In particular, the case packer group is responsible for putting the blisters into the case and sticking its edges. The incorrect execution of the sticking phase may generate unacceptable final products. For this reason, it is necessary to recognize the malfunction as soon as possible. To this aim, the current (mA) absorbed by the main actuator of the system is collected with a sampling frequency of 500 Hz under different conditions. In total, one normal condition and four fault conditions were simulated, i.e., without load, with overload, with a regular load but without withdrawal of the die-cuts, and the emptying of the last die-cuts. The dataset collected in the nominal condition and in the first three improper conditions include 100 working cycles. The dataset corresponding to the last fault condition includes only 47 working cycles. Each cycle is made of 75 data samples and lasts 0.15 s. Therefore, 15 s of data have been collected for each condition, except for fault 4, for which the monitoring is 7.05 s. Figure 2 shows the trend of the current signal during one cycle for each condition.

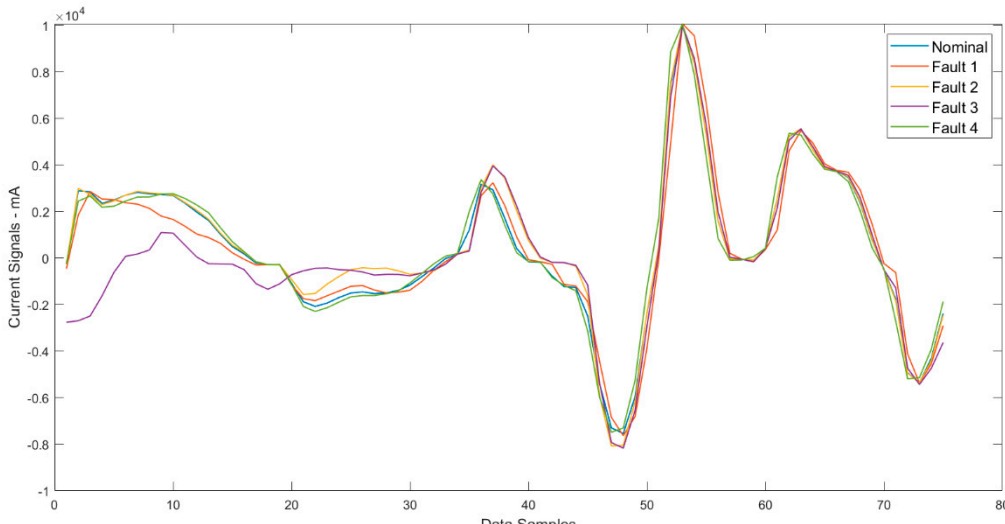

**Figure 2.** Current signal trends during the nominal condition and the four improper working conditions.

*4.3. Dataset 3: Suction Cups*

Like the second dataset, this dataset refers to a critical part of a packaging machine whose malfunction strongly affects the products' quality. The component under analysis consists of four suction cups that push the cartoon forward by rotating. A full cycle lasts 0.134 s. In a cycle, the pressure of the suction cups on the cartoon may decrease, causing the detachment of one or more suction cups. If two suction cups detach, the component cannot perform its function correctly and an intervention is required. The detachment of the suction cup is almost instantaneous. A pressure sensor is, therefore, placed on the component to detect the pressure drops corresponding to the detachment of the suction cups. Signals are collected at a sampling frequency of 10 kHz, and three experiments are conducted. First, the data are collected under a nominal condition; then, two fault conditions are generated, having one (Fault s1) and two (Fault s2) suction cups detached, respectively. The pressure values in the different conditions are shown in Figure 3.

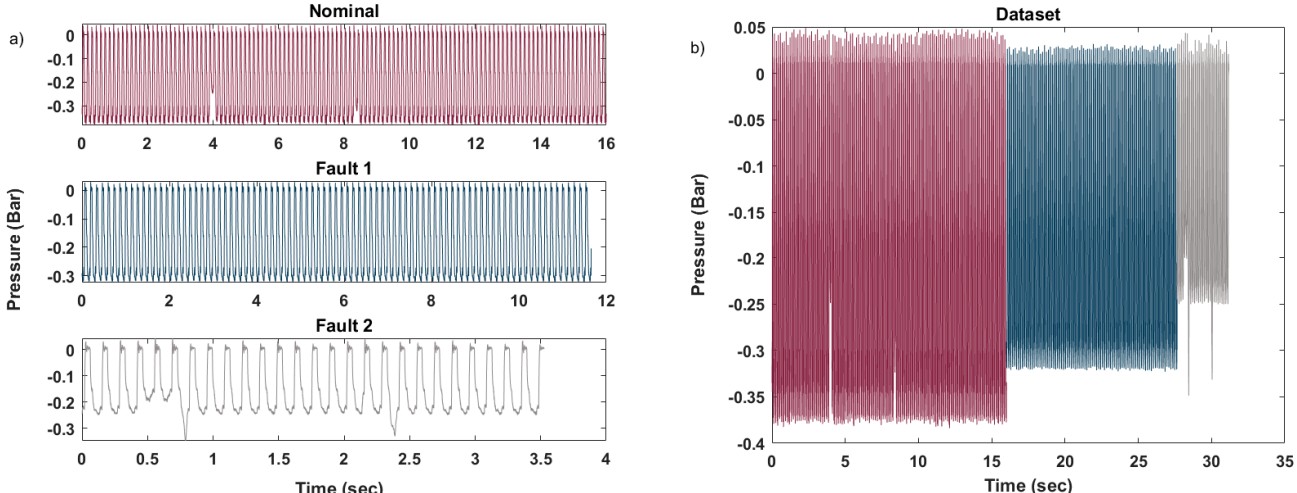

**Figure 3.** Pressure signals produced by the suction cups during the nominal condition and the two fault conditions, (**a**) considered separately, and (**b**) put in sequence to simulate the detachment of the first and second suction cup.

*4.4. Dataset 4: Extruder*

This case study analyzes the behavior of an automatic machine based on signals collected from 30 different sensors placed on a subgroup made of two electric motors,

a heated extruder, and a volumetric pump. The sensors measure the temperature and the percentage of usage of 9 thermos-resistors in different zones, the motor's power and velocity, and the input and output pressures of a volumetric pump placed at the end of the extruder. The operating condition is determined by a set of temperature values, whose actual signals are collected during the machinery's functioning. Signals are collected from two different sources. In the first one, signals are acquired at a frequency of 1Hz. Each day, almost 8 h of data are recorded; then, files are directly downloaded by the PLC of machinery. The second one is a low-frequency data source: each day, after 30 min of the machine functioning, four different statistics over a batch of 30 s for each collected signal are computed.

The available dataset includes data related to two different machines, M1 and M2. M1 was monitored for almost 11 months, while M2 was observed for nearly 21 months. For each machine, two different settings were implemented. While, for M1, there was only one change from setting 1 to setting 2, M2 worked under setting 1, setting 2, and again under setting 1. Figure 4 shows the raw temperature signals of M1 and M2.

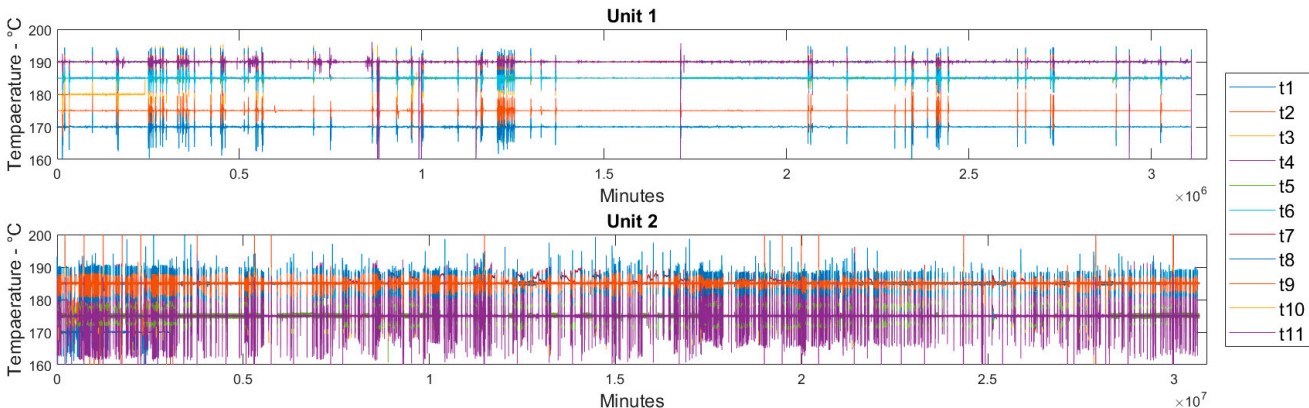

**Figure 4.** Temperature signals collected during the two nominal conditions by M1 (on the **top**) and M2 (on the **bottom**).

Although many data were available, this case study only considers the data for which the operating condition can be extracted from the low-frequency data source.

### 4.5. Dataset 5: Sealing Group

The fourth dataset is collected from a thermoforming machine that can produce up to 320 products per minute. In particular, a component-level analysis is conducted on the sealing group, which is responsible for sealing the upper and bottom parts of the product. The group is made of two tables in which the film flows continuously, allowing the sealing at a low temperature. In each cycle, the top table goes up and closes; the cover is sealed to the base, and the tables open before returning to the initial position. The health condition of the sealing group is monitored through a displacement sensor that records the positions of the two tables. Figure 5 depicts the trend of the measured signal during a sealing cycle, which lasts 800 milliseconds, and the sampling frequency of the sensor is 1000 Hz. One nominal condition and three unacceptable working conditions, nominal, fault 1, fault 2, and fault 3, are available. Fault conditions do not cause the system breakdown. However, the quality of the final product is compromised, and the improper working condition has to be detected as soon as possible to avoid production losses.

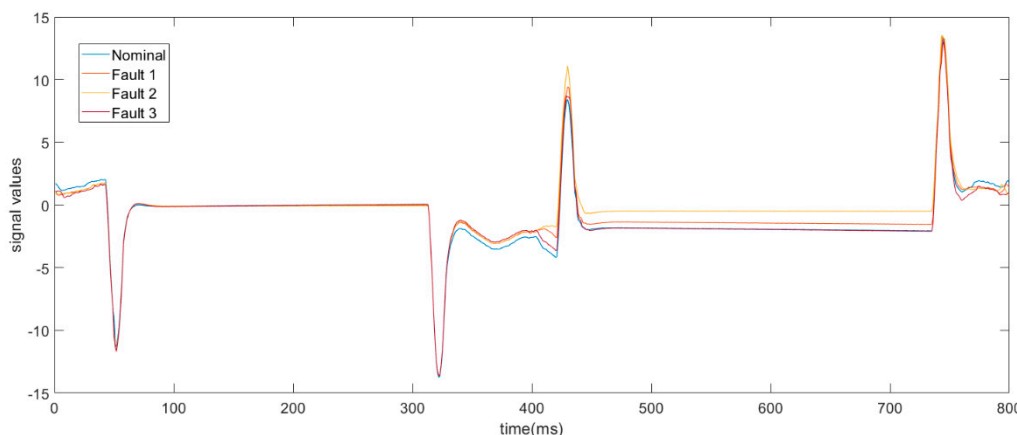

**Figure 5.** Raw signals produced by the sealing group during the nominal condition and the three fault conditions.

*4.6. Experimental Scenario*

For each case study, two main tasks are carried out: feature extraction and fault/novelty detection. Because features have to be extracted from streaming data in real time, only time-domain features have been considered. In particular, the peak, peak-to-peak, mean, root mean square, crest factor, kurtosis, skewness, shape factor, and impulse factor are computed over batches of different lengths depending on the specific case. Then, two methods for feature selection and learning, i.e., the Pearson correlation analysis (PC) and the principal component analysis (PCA) have been used to reduce the dimensionality of the dataset. Finally, the AAD and ADP algorithms for anomaly and novelty detection are applied from scratch or after a pretraining phase to detect novel faults (improper conditions) or novel machinery settings. The mathematical background of feature extraction and selection (PC and PCA) methods and the clustering-based novelty detection approach (AAD + ADP) used in this paper are provided in Appendices A and B, respectively.

In particular, two approaches have been adopted for different case studies, i.e., clustering-based with pretraining and clustering-based from scratch. The schemes of the two approaches are depicted in Figure 6.

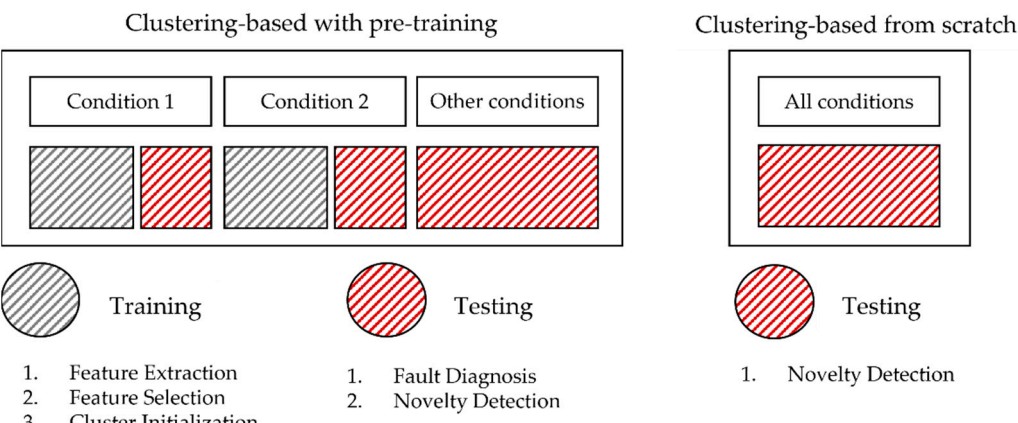

**Figure 6.** Experimental scenario.

In the clustering-based with pretraining approach, the dataset is divided into two sets, named training and testing sets, including 70% and 30% of observations, respectively. The training set is first used to extract and select the relevant features revealing the health condition of monitored components. Then, the AAD + ADP (described in Appendix B) is applied to the selected features, and the local parameters of created clusters, i.e., the number of observations and the clouds' centers, are stored. Finally, the remaining observations

are used to test the effectiveness of the algorithm to (1) assign the 30% of observations corresponding to the conditions included in the training test to the correct clusters among the initialized ones and (2) to detect the change towards a novel condition and create a new cluster.

In the clustering-based from scratch approach, no cluster is initialized. The algorithm starts from the first data creating a cluster; then, it creates a new cluster each time it recognizes a novel condition.

Note that the testing phase of the approaches is applied to streaming data. Hence, the algorithm reads data batches of a certain length, extracts one or more features, and creates a label corresponding to the assigned cluster.

According to the framework proposed in [19], novelty detection is also applied at a system level to discover novel operating conditions of the machinery. Therefore, some of the case studies deal with the operating recognition problem.

Table 2 summarizes the goal (fault detection or operating condition recognition), the approach, the method, and the feature engineering method adopted for each case study.

**Table 2.** Description of each case study's goals, approaches, and feature extraction methods.

| Dataset | Goal | Approach | Feature Engineering Method |
|---|---|---|---|
| Electric motor | Operating Condition recognition and fault detection | Clustering-based with pretraining | Manually selected features |
| Case packer | Fault detection | Clustering-based from scratch | Pearson Correlation, Principal Component Analysis |
| Suction cups | Fault detection | Clustering-based from scratch | Pearson Correlation |
| Automatic Machinery | Operating Condition recognition | Clustering-based from scratch | Principal Component Analysis |
| Sealing group (displacement) | Fault detection and diagnosis | Clustering-based with pretraining | Pearson Correlation Analysis |

## 5. Results

This section summarizes the results obtained by applying the AAD + ADP method to the five datasets described in previous sections.

In particular, results are provided in terms of the number of observations correctly assigned to the clusters, the latency of the detection, and the number of false alarms. The number of false alarms is the number of points recognized as anomalies that do not cause a cluster change. In other words, false alarms are those anomalies that are not caused by the occurrence of a failure or a modification of the setting.

### 5.1. Dataset 1: Electric Motor

In this case, the clustering-based with the pretraining approach is used. In particular, 70% of observations corresponding to condition 1 and condition 2 are included in the training set. The goal is twofold. First, the analysis at a system level is conducted to recognize known and novel system operating conditions; second, the analysis at a component level is conducted to detect the fault of the electric motor.

To achieve these aims, time-domain features are extracted from signal batches of 1 s (12.8 samples). Then, the most relevant time-domain features are selected according to the following criteria: system-level features have to be as similar as possible in the same operating condition and as distinct as possible among different operating conditions; component-level features have to be as constant as possible during the two nominal conditions. The selected system-level and component-level features are shown in Figure 7. At a component level, the mean of three different signals (sensor 2, x-axis and z-axis, and sensor 3, y-axis) is selected, while the impulse factor of sensor 3, z-axis, is selected at a component level.

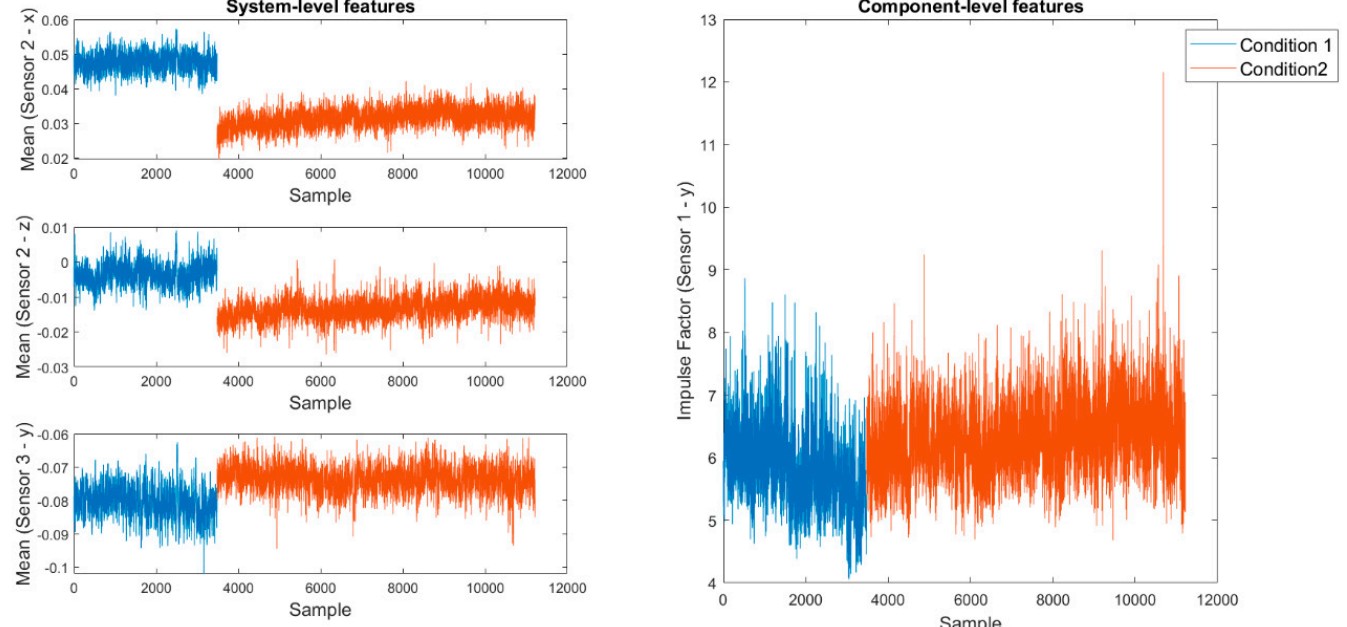

**Figure 7.** Selected system-level features (**left**) and component-level features (**right**).

Then, the ADP algorithm is applied to find the clusters' corresponding local parameters, whose results are shown in Table 3.

**Table 3.** Cluster local parameters obtained from the pretraining phase.

| Parameter | Operating Condition Recognition | | Fault Detection |
|---|---|---|---|
| Number of clusters | 2 | | 1 |
| Number of points | 3493 | 7722 | 11,215 |
| Cluster center | (0.0476; −0.0034; −0.0805) | (0.0315; −0.0134; −0.0736) | 6.1358 |

Then, the AAD + ADP algorithm is applied to all observations of each feature set. For the operating condition recognition task, streaming novelty detection aims to create two new clusters when condition 3 and the fault to the electric motor occur. The goal of the fault detection task is to create a new cluster when the fault occurs. Results are shown in Figure 8, and the algorithm's performances are summarized in Table 4.

For the operating condition recognition, the points corresponding to conditions 1 and 2 are assigned to the initialized clusters correctly. The algorithm detects the change from condition 1 to condition 2 after 15 s and to condition 3 after 10 s. However, the fourth condition is not detected by the selected features. Concerning the fault detection task, the algorithm creates a novel cluster 18 s after the change from condition 3 to condition 4 (yellow dots in Figure 8b); then, a second cluster is created after 175 s, which corresponds to almost 23 min before the motor failure. Therefore, the system breakdown could have been avoided if an alarm was triggered at the second cluster creation time. In addition, all points assigned to cluster 2 can be labeled and used to update classification models for diagnostics.

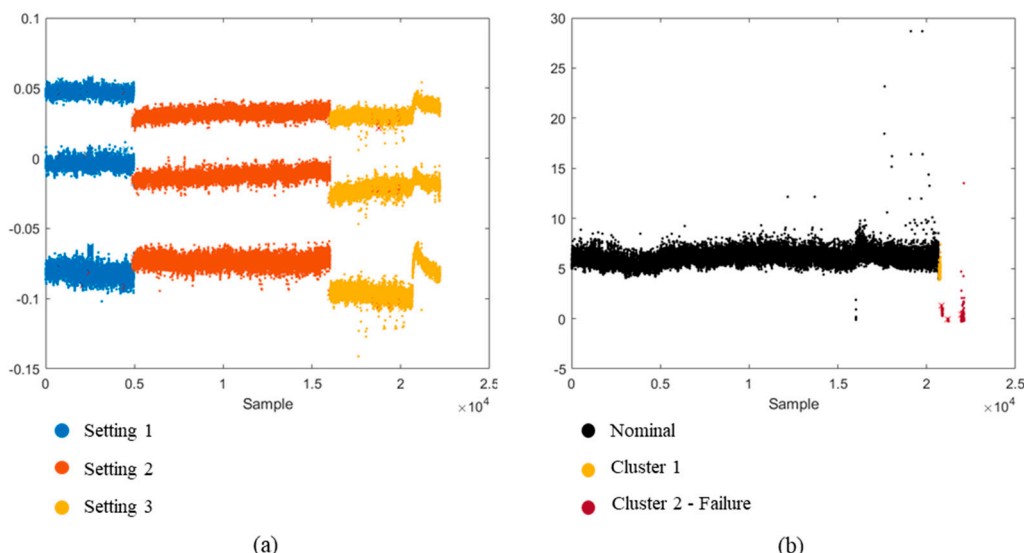

**Figure 8.** Results of the AAD + ADP algorithm (electric motor) for (**a**) operating condition recognition, and (**b**) fault detection.

**Table 4.** Performances of clustering-based with pretraining approach (electric motor).

| Condition | Detected Setting Change (Sample) | Latency (s) | N Points Assigned to the Corresponding Cluster | False Alarms |
|---|---|---|---|---|
| Setting 1 | - | | 4982 | |
| Setting 2 | 4.984 | 15 | 11,048 | |
| Setting 3 | 16.032 | 10 | 6187 | 168 |
| Setting 4 | - | - | - | |
| | | | 20,681 | |
| Motor Fault | 20,682 | −1536 | 157 | 488 |
| | 20,839 | −1379 | 1380 | |

### 5.2. Dataset 2: Case Packer

In this case, the main goal is to detect a known fault as soon as possible, without any training phase. Faults have been separately considered since they cannot occur in sequence. Hence, four scenarios have been simulated, in which the dataset corresponding to one fault is appended after the nominal behavior. In this case, two distinct feature sets are considered: the first one, which includes the peak, the mean, the root mean square, and the crest factor, has been obtained through the Pearson correlation analysis. The second feature set includes the principal components (PCs) obtained through the PCA.

Results of the AAD + ADP algorithm applied to the two sets of features, i.e., features selected through the Pearson correlation analysis and the features extracted through the PCA, are shown in Figure 9a and Figure 9b, respectively. This figure depicts the points assigned to different clusters in different colors and anomalies with red crosses. As can be seen in Figure 9b, the extracted PCs are not able to reveal the health condition of the component. In contrast, the performances of the algorithm applied to the features selected through the Pearson correlation analysis are summarized in Table 5. In all cases, a false positive (red cross) is detected during the nominal behavior. Fault 1, Fault 2, and Fault 3 are detected after five cycles (0.75 s) of their occurrence, and a novel cluster is created when the anomaly corresponding to the fault is detected. Instead, when Fault 4 occurs, an anomaly is detected. However, the following points continue to be assigned to the cluster corresponding to the nominal behavior.

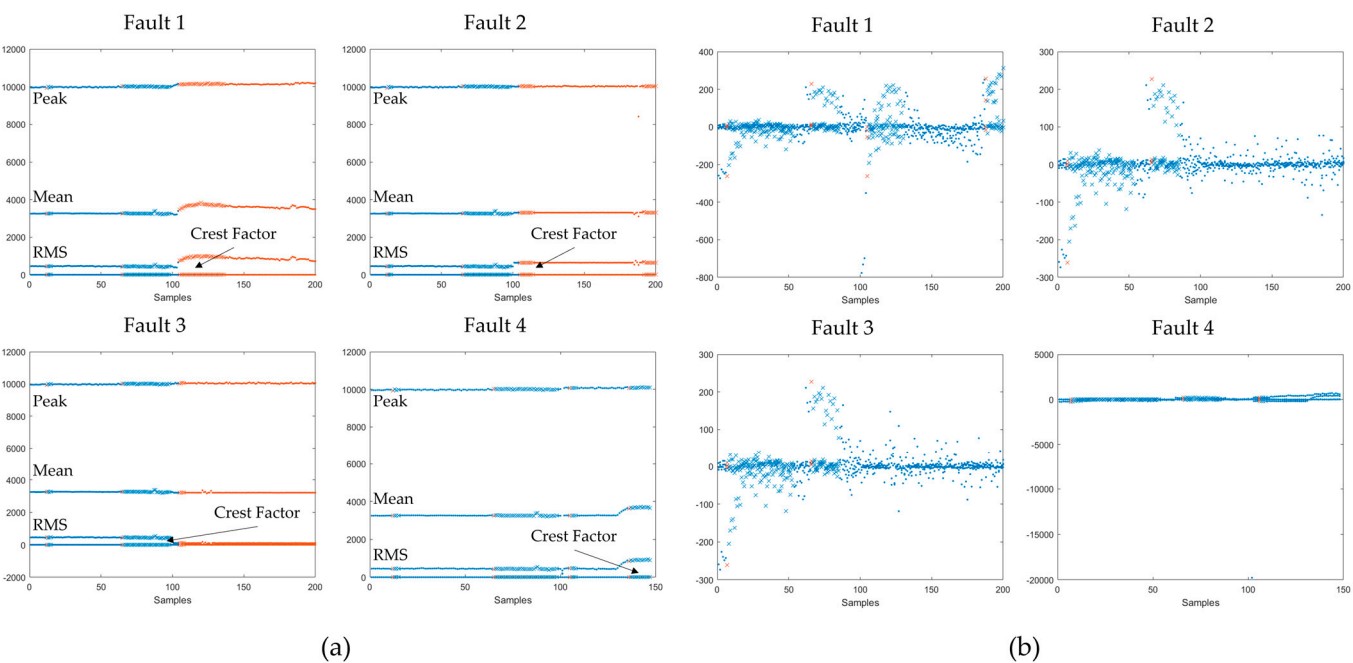

(a)

(b)

**Figure 9.** Results of the AAD + ADP algorithm applied to (**a**) features selected through the Pearson correlation analysis and (**b**) features extracted through the PCA (case packer).

**Table 5.** Performances of the clustering-based from scratch approach (case packer).

| Fault | Cycle of Detection | Latency (s) | N Created Clusters | N Points | False Alarms |
|---|---|---|---|---|---|
| Fault 1 | 105 | 0.75 | 2 | Cluster 1 = 104 Cluster 2 = 96 | 69 |
| Fault 2 | 105 | 0.75 | 2 | Cluster 1 = 104 Cluster 2 = 96 | 56 |
| Fault 3 | 105 | 0.75 | 2 | Cluster 1 = 104 Cluster 2 = 96 | 41 |
| Fault 4 | - | | 1 | Cluster1 = 147 | 54 |

### 5.3. Dataset 3: Suction Cups

The goal of this case study is to detect the occurrence of a failure as soon as possible. Unlike the case packer case study, in which the available fault conditions could not occur in sequence, in this case, the two available fault conditions correspond to two different severities of the same fault mode. Hence, the AAD + ADP algorithm is applied from scratch to signals corresponding to all conditions. The peak, the peak-to-peak, the mean, and the RMS are selected through the Pearson correlation analysis used as the input of the streaming algorithm. Results are shown in Figure 10 and summarized in Table 6. The first behavior change is detected after five cycles, equal to 0.67 s, while the second change is detected as it occurs, with zero latency. For each condition, a cluster is created and points are correctly assigned to the clusters. In addition, 55 false alarms have been avoided.

**Table 6.** Performances of the clustering-based from scratch approach (suction cups).

| Fault | Cycle of Detection | Latency (s) | N Created Clusters | N Points | False Alarms |
|---|---|---|---|---|---|
| Fault s1 | 124 | 0.67 | 1 | 88 | 55 |
| Fault s2 | 212 | 0 | 1 | 21 | |

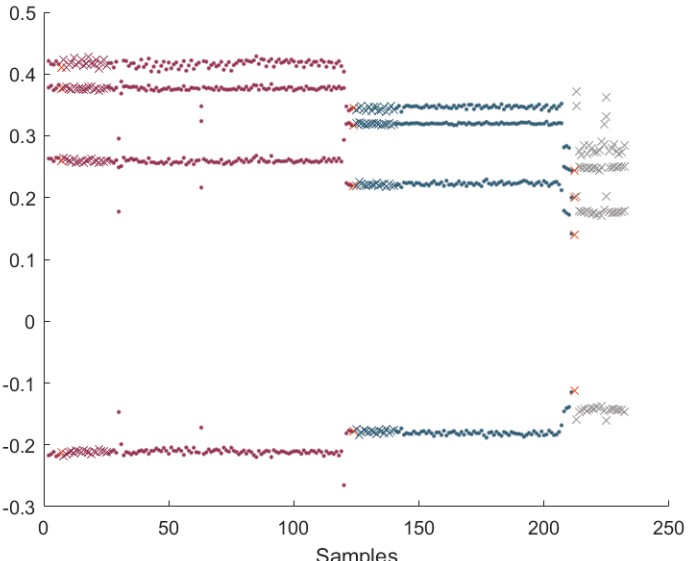

**Figure 10.** Results of the AAD + ADP algorithm (suction cups).

### 5.4. Dataset 4: Automatic Machinery

In this case, the AAD + ADP approach is applied from scratch to the extracted PCs. As summarized in Table 7, in both cases, the algorithm recognizes the change from setting 1 to setting 2 after nine data samples. In addition, the algorithm identifies the switch from setting 2 to setting 1 for M2, as data points are assigned to the existing cluster corresponding to the first operating condition. Results are also shown in Figures 11 and 12, where the black dots correspond to the first condition, the grey dots to the second condition, and the red crosses represent the moment in which a changing behavior is detected.

**Table 7.** Performances of the clustering-based from scratch approach (extruder).

| Unit | Setting Change (Detected) | Setting Change (Actual) |
|------|---------------------------|-------------------------|
| M1 | 114 | 105 |
| M2 | 4257 | 4248 |
| | 5674 | 4904 |

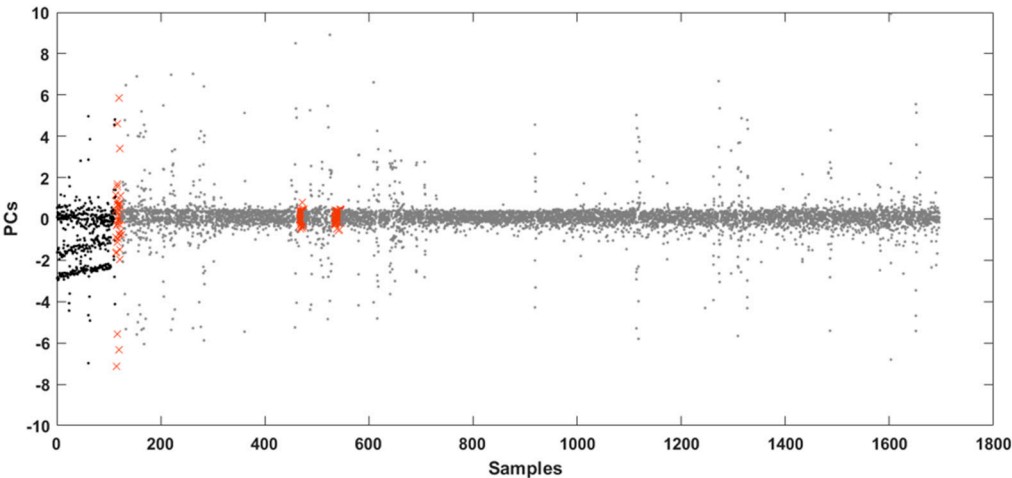

**Figure 11.** Results of the AAD + ADP algorithm (extruder-M1).

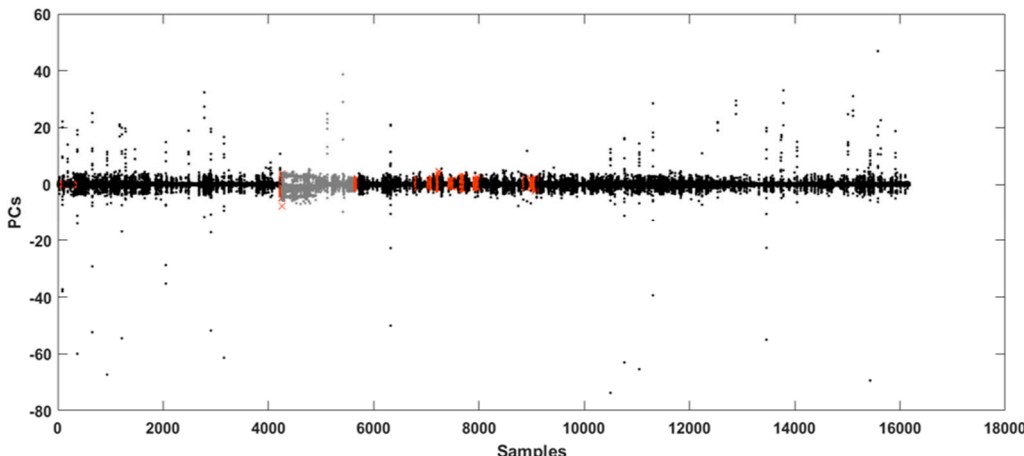

**Figure 12.** Results of the AAD + ADP algorithm (extruder-M2).

Note that, in both cases, there are several data points considered anomalous. However, none of them creates a new cluster. These points correspond to true anomalies in the data, such as measurement errors or anomalous peaks, which are evident in the raw signals.

*5.5. Dataset 5: Sealing Group*

In this case study, the AAD + ADP with pretraining has been used. Similar to the case study shown in Section 5.2, the three fault conditions correspond to three distinct improper working conditions. Hence, the algorithm is applied separately for each condition. The pretraining consists of selecting the most relevant features using 70% of the observations corresponding to the nominal condition and the condition fault 1. These data are also used to initialize the local parameters of the clusters with the ADP algorithm. Then, the remaining observations are used to demonstrate that the algorithm assigns the points belonging to the nominal condition and the condition fault 1 to the corresponding existing clusters and creates new clusters for faults 2 and 3.

Features selected through the Pearson correlation analysis are shown in Figure 13, while the local parameters of the clusters obtained during the training phase are summarized in Table 8.

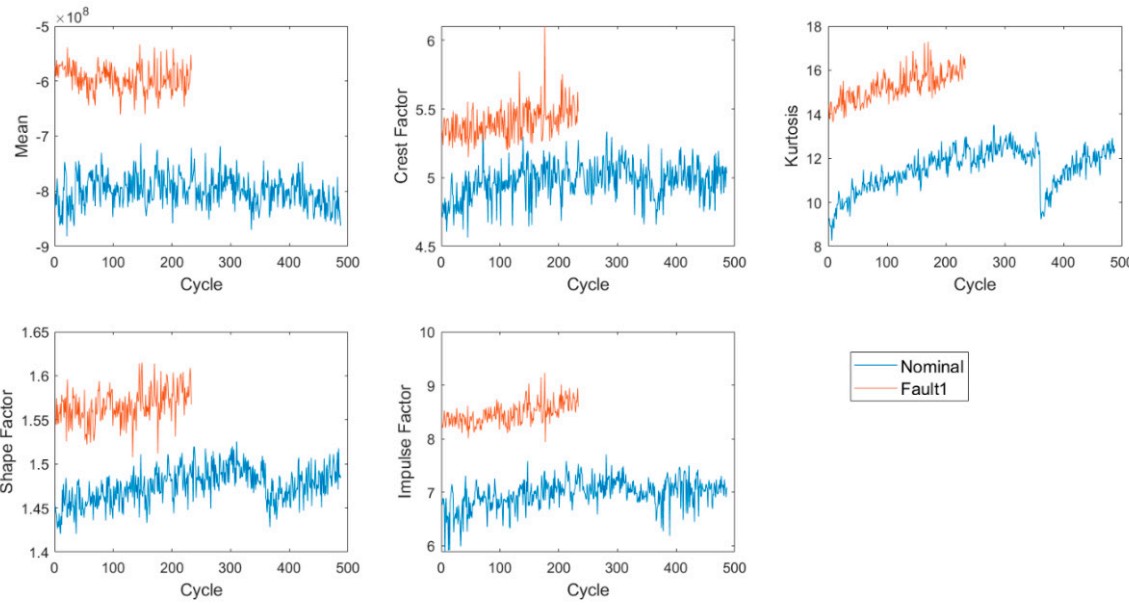

**Figure 13.** Selected features (sealing group).

**Table 8.** Cluster local parameter obtained from the pretraining phase (sealing group).

| Parameter | Nominal–Fault 1 | |
|---|---|---|
| Number of clusters | 2 | |
| Number of points | 493 | 297 |
| Cluster center | $(-7.6941 \times 10^8; 4.9795;$ <br> $11.5270; 1.4761; 6.993)$ | $(-5.9629 \times 10^8; 5.4121;$ <br> $15.2841; 1.5666; 8.4756)$ |

Finally, the algorithm is applied to all observations of each condition. The performances of the algorithm applied to the features selected through the Pearson correlation analysis are summarized in Table 9. In the first case, when the nominal condition and fault 1 are considered, fault 1 has been detected with a latency of 10 cycles (8 s). In addition, no cluster is created, and the points are correctly assigned to cluster 1 and cluster 2. Similarly, fault 2 is detected after 8 s, and the points corresponding to the nominal condition are correctly assigned to cluster 1. Furthermore, a new cluster is created when fault 2 occurs. Finally, fault 3 is not detected by the algorithm. Figure 14 shows the results of the AAD + ADP algorithm for each condition

**Table 9.** Performances of AAD + ADP (sealing group).

| Conditions | Cycle of Detection | Latency (Sample) | N Created Clusters | N Points | False Alarms |
|---|---|---|---|---|---|
| Nominal-Fault 1 | 706 | 10 | 0 | Cluster 1 = 705 <br> Cluster 2 = 324 | 16 |
| Nominal-Fault 2 | 706 | 10 | 1 | Cluster 1 = 705 <br> Cluster 2 = 191 | 13 |
| Nominal-Fault 3 | - | - | 0 | Cluster 1 = 886 | 10 |

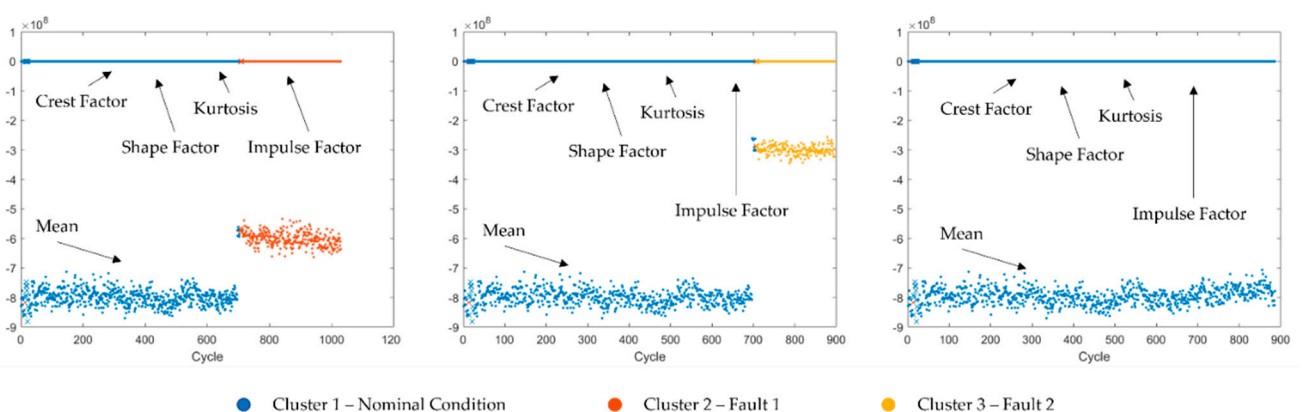

**Figure 14.** Results of the AAD + ADP algorithm (sealing group).

## 6. Discussion and Conclusions

In this paper, a method for data reduction, collection, and labeling is validated on five datasets collected from the industrial field. The main considerations will be described at two levels: the data analysis method and the data and information collection framework.

At the analysis level, two primary considerations can be drawn. First, the features extracted in the time domain and selected through the Pearson correlation analysis, which reduces the redundancy of information included in the features by selecting only the less correlated ones, are able to reveal all health and fault conditions, except for one fault condition of the case packer and one fault condition of the sealing group. Therefore, these methods can properly extract and select relevant information from raw data, even if few data are available for each condition. In contrast, if a fault condition is not included in the training set, it may happen that the selected features cannot distinguish it from the nominal

behavior. However, in a previous paper [58], the authors demonstrated that the fault condition of the sealing group not detected by the AAD + ADP approach is not recognized by a classification-based approach either. Therefore, the issue is related to the collected signal, which cannot provide information about that improper condition.

The second consideration at the analysis level concerns the application of the streaming AAD + ADP algorithm. This approach effectively recognizes known and novel behaviors with low latency in all case studies. In addition, it is fast and requires little computation effort.

At the framework level, the description of the datasets provided by industries and the application of the streaming clustering-based approach for fault and novelty detection led to two main considerations.

The first consideration concerns the characteristics of the available datasets. As can be seen in Table 1, the datasets are significantly unbalanced. In particular, when the electric motor failed, the system broke down and only 25.8 min of fault signals could be collected. In contrast, 344.4 min of signals are collected during the nominal operating conditions. Similarly, the detachment of the second section cups in one of the monitored automatic machinery led to an acceptable working condition; therefore, only 3.52 s of signals were collected. In the case study introduced in Section 5.4, aimed at recognizing the operating condition of the system, it seems that no faulty data are available at all. However, the data cover a long period (from 1 to 2 years), and several improper working conditions occurred during those periods. The data collection problems, in this case, concern the lack of a label and the quantity of data. Because the machine generates many gigabytes of data, it cannot be monitored continuously. Therefore, only some batches of data per day are collected, which are unlabeled and without any contextual information. This consideration implies a huge effort for the data analysts and makes the data-driven approach less effective.

The second consideration concerns the possibility of solving the above issues by applying a streaming approach at the edge of the machinery to reduce the quantity of data to transfer, store, and automatically label the collected data. Indeed, raw signals can be processed at the edge to extract relevant features. All case studies show that a set of features selected by considering only some health conditions can also reveal other conditions. Therefore, it is not necessary to permanently store all raw signals. On the contrary, only features can be stored, which can be used to construct more complex models. In this way, the data quantity is notably reduced, and machinery can be monitored continuously. In addition, real-time clustering allows for labeling the data during the collection. Because the algorithm is incremental, the data corresponding to a novel condition are assigned to the same cluster and will have the same label. From the industrial point of view, obtaining the label in real time allows for detection of faults or anomalous behaviors as they occur, and, therefore, it is possible to intervene before more severe faults occur and the system breaks down.

In conclusion, using the proposed approach, it is possible to implement an edge-cloud infrastructure to deal with data availability, completeness, and quantity issues. In particular, feature extraction can be extracted in real time to reduce the quantity of data to analyze. In addition, fault and novelty detection approaches should be applied to streaming data to obtain information on the setting, which corresponds to the operating condition of the system, and the event, which corresponds to the occurrence of a particular event, such as a fault, a novel behavior, or a change in the system setting. This information is provided in real time to have feedback on the health condition of the machinery; at the same time, it is included in the dataset so that the extracted file can be directly used as input for diagnostics models without any preprocessing phase.

**Author Contributions:** Conceptualization and methodology, F.C. and A.R.; software development, F.C.; experimental dataset collection, A.R.; new approach application, F.C. and F.G.G.; results analysis and discussion. F.C. and M.B.; writing—original draft preparation, F.C.; writing—review and editing, A.R. and M.B. All authors have read and agreed to the published version of the manuscript.

**Funding:** This research received no external funding.

**Data Availability Statement:** Restrictions apply to the availability of these data. Data were obtained from Authors (lab dataset) and third parties (industrial datasets) and are available from the authors with the permission of third parties (industrial datasets).

**Conflicts of Interest:** The authors declare no conflict of interest.

## Appendix A

The formulas of the time-domain features typically extracted from condition monitoring data and also used in this paper are provided in Table A1.

**Table A1.** Extracted time features from raw signals. $x_i$, is the signal (observation) at the time $i$, $i = 1 \ldots S_N$, $S_N$ is the length of the signal segments, $\bar{x}$ is the mean value of $x_i$, $i = 1 \ldots S_N$, and $\sigma^3$ $\sigma^4$ are third and the fourth moment of $x_i$, $i = 1 \ldots S_N$, respectively.

| Feature Name | Formula |
|---|---|
| Peak | $f_{Peak} = x_{max} = max|x_i|$ |
| Peak-to-peak | $f_{Peak2Peak} = |max(x_i) - min(x_i)|$ |
| Mean | $f_{Mean} = \frac{1}{N} \sum_{i=1}^{N} x_i$ |
| Root Maen Square (RMS) | $f_{RMS} = \sqrt{\frac{1}{N} \sum_{i=1}^{N} x_i^2}$ |
| Crest Factor (CF) | $f_{CrestF} = f_{Peak} / f_{RMS}$ |
| Kurtosis | $f_{Kurt} = \frac{1}{N} \sum_{i=1}^{N} \frac{(x_i - \bar{x})^4}{\sigma^4}$ |
| Skewness | $f_{Skew} = \frac{1}{N} \sum_{i=1}^{N} \frac{(x_i - \bar{x})^3}{\sigma^3}$ |
| Shape Factor | $f_{ShapeF} = f_{RMS} / f_{Mean}$ |
| Impulse Factor | $f_{ImpulseF} = f_{Peak} / f_{Mean}$ |

Pearson correlation analysis is a filter-based method that considers variables with a high linear relationship containing redundant information [59]. The level of correlation between two features is usually provided by a fixed threshold, which has been set equal to the absolute value of 0.9 in this paper. Then, the Pearson correlation coefficient is computed among all features through Equation (A1) to build the correlation matrix.

$$r_{XY} = \frac{Cov(X,Y)}{\sqrt{Var(X)Var(Y)}} \tag{A1}$$

where $X$ and $Y$ are two random variables, $Cov(X,Y)$ is the covariance of the two variables, and $Var(X)$ and $Var(Y)$ are the variance of the variable $X$ and $Y$, respectively. The Pearson correlation coefficient, $r_{XY}$, is always included in the range $[-1, 1]$, where $-1$ indicates that the two variables are negatively correlated, 1 indicates that the variables are positively correlated, and 0 that the variables do not correlate. After the correlation matrix construction, an iterative process for feature elimination is conducted. Hence, for each column, and for each row, if $r_{XY} \leq -0.9$ or $r_{XY} \geq 0.9$, the column is eliminated.

The principal component analysis (PCA) [60] is a feature learning method. It is a statistical analysis approach to mapping multiple characteristic parameters to a few comprehensive features. These PCA-based comprehensive features are not related to each other and can represent original fault features effectively [61]. Given a dataset $X$, of dimension $m$, PCA aims to find a set of orthonormal basis vectors of dimension $p < m$, which are called principal components (PCs), that maximize the variance over the dataset when it is projected onto the subspace spanned by these PCs. Basically, if we have data points in a two-dimensional space and we want to project them in one-dimensional space, what PCA does is to find the direction of the vector and the position of the points on that vector, which is expressed by coefficients, such that the reconstruction or projection error is minimized. To this aim, the covariance matrix of the dataset is first computed, and the eigenvalues and eigenvectors are extracted. The eigenvectors correspond to the PCs, while

the corresponding eigenvalues represent the variance associated with that PC. The PCs are selected so that the cumulative variance described by them is higher than a certain percentage (usually from 90 to 99%).

**Appendix B**

Angelov, Ramezani, & Zhou [62] introduced the concept of recursive density estimation (RDE) in the context of detection and object tracking in video streams. Aiming to decide whether a pixel belongs to the background or the foreground in real time, the introduced approach substitutes the standard Gaussian kernel adopted for modeling the pixel probability density function (pdf) with the Cauchy function, which allows the recursive estimation of pixel pdf as a new image frame occurs. In Costa et al. (2015), concepts of RDE theory are adopted in the context of online fault detection in order to discover anomalous behaviors. The parameters involved are the global density, the mean value, and the scalar product, which can be recursively computed by Equation (A2), Equation (A3), and Equation (A4), respectively.

$$D(x_k) = \frac{1}{1 + \|x_k - \mu_k\|^2 + \Sigma_k - \|\mu_k\|^2} \tag{A2}$$

$$\mu_k = \frac{k-1}{k}\mu_k + \frac{1}{k}x_k \tag{A3}$$

$$\Sigma_k = \frac{k-1}{k}\Sigma_k + \frac{1}{k}\|x_k\|^2 \tag{A4}$$

where $x_k \in \mathbb{R}^n$ is the feature vector at the time stamp $k$.

At the first iteration $k = 1$, the parameters are initialized as $D(x_1) = 1$, $\mu_1 = x_1$, $\Sigma_1 = \|x_1\|^2$. Then, for each $k > 1$, the parameters are updated, and the condition expressed in Equation (A5) is checked to decide whether the current point represents an anomaly or not:

$$IF\ D(x_k) < \mu_D\ for\ k = t_1, \ldots, k-1, k\quad THEN\ x_k\ is\ an\ anomaly \tag{A5}$$

where:

$$\mu_D = \left(\frac{ks-1}{ks}\mu_D + \frac{1}{ks}D(x_k)\right)(1 - \Delta_D) + D(x_k)\Delta_D \tag{A6}$$

is the mean value of the local density computed recursively, $\Delta_D = |D(x_k) - D(x_{k-1})|$ is the absolute value of the difference between the global density computed at two consecutive time stamps, and $ks$ is the number of data samples from the last status change. Indeed, if the condition is satisfied, then the status of the system switches from normal to anomalous, and $ks$ is set to 0. The condition expressed in Equation (A5) means that, if the global density is lower than the mean density for a certain number of time stamps (or seconds), the status becomes anomalous. Indeed, when a new data point arrives, if it is close to the previous point, then $\mu_k$ is close to the $x_k$, $D(x_k)$ stays close to 1, and $\mu_D$ stays close to the actual mean of the data points, as $(1 - \Delta_D)$ is very close to 1, which gives more importance to the first term of Equation (A6). However, when the new point is far from the previous ones, $D(x_k)$ slightly decreases, while $\mu_D$ becomes closer to $D(x_k)$, as the term $\Delta_D$ gives more importance to the second term of Equation (A6). As new points are closer to the previous point, then $D(x_k)$ continues to decrease until the condition in Equation (A5) is satisfied. Note that, when the status changes from normal to anomalous, $ks$ is set to 0, leading $\mu_D$ to notably decrease. When the mean density is lower than the global density for a certain number of points, or seconds, the status returns to the normal condition. Thus, the condition expressed by Equation (A7) applies:

$$IF\ D(x_k) > \mu_D\ for\ k = t_2, \ldots, k-1, k\quad THEN\ x_k\ is\ normal \tag{A7}$$

Note that both $t_1$ in Equation (A5) and $t_2$ in Equation (A7) are set by the user and may represent either the data samples or seconds.

Based on the concepts of RDE, a clustering algorithm has also been introduced in [54], in both offline and online versions. Here, the online version will be briefly described. At the first iteration ($k = 1$) the local parameters of each cluster are initialized as follows:

$$C_k = 1; \ \mu_k^1 = x_1; \ S_k^1 = 1 \tag{A8}$$

where $C_k$ is the cluster at the time stamp $k = 1$, $\mu_k^1$ is the focal point of cluster 1 at the time stamp $k = 1$, and $S_k^1$ is the number of data points belonging to cluster 1 at the time stamp $k = 1$. In addition, the parameters expressed by Equations (A3) and (A4) are also initialized. Then, for each $k > 1$.

The mean value $\mu_k$ and the scalar product $\Sigma_k$ are updated by means of Equation (A3) and Equation (A4), respectively.

The condition expressed by Equation (A9) is checked to decide whether the current point has to be assigned to an existing cluster or should be a new focal point itself:

$$IF \ D(x_k) > \max_{i=1,\dots,C_k} D_k(\mu_k^i) \ OR \ D(x_k) < \min_{i=1,\dots,C_k} D_k(\mu_k^i)$$
$$THEN \ x_k \ becomes \ a \ new \ focal \ point \tag{A9}$$

The condition expressed in Equation (A9) means that, if the global density is higher than or lower than the densities computed at each of the existing focal points (the density of each cluster), then the current point creates a new cluster.

If Equation (A9) is satisfied, then a new cluster is created whose parameters are initialized by means of Equation (A10):

$$C_k = C_{k+1}; \ \mu_k^{C_k} = x_k; \ S_k^{C_k} = 1 \tag{A10}$$

Otherwise, the distance between the current feature vector $x_k$ and the focal point of each existing cluster is computed in order to decide whether the current point can be assigned to the closest cluster, using Equation (A11):

$$IF \ \|x_k - \mu_k^n\| < \sqrt{\Sigma_k - \|\mu_k\|^2}$$
$$THEN \ x_k \ is \ assigned \ to \ \mu_k^n \tag{A11}$$

The condition expressed by Equation (A11) means that, if the distance between the current point and the nearest focal point $\mu_k^n$ is lower than the distance among all points arrived until the current timestamp, the current point is assigned to the nearest cluster.

If Equation (A11) is satisfied, then the parameters of the closest cluster $C_k^n$ are updated through Equation (A12):

$$\mu_k^n = \frac{S_k^n - 1}{S_k^n} \mu_k^n + \frac{1}{S_k^n} x_k; \ S_k^n = S_k^n + 1 \tag{A12}$$

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
