# Peer review of "Data-Driven Fault Detection and Diagnosis: Challenges and Opportunities in Real-World Scenarios"

_applsci, doi:10.3390/app12189212_

Round 1

Reviewer 1 Report

Very interesting work.

Comments to the authors:

1-      In the section Introduction: the review of the present work is insufficient, which in turn makes the research weak. Other literature on numerical models for the same purpose (or similar) can be added, addressing the assumptions, main challenges and advancements. In the published literature, there are several approaches (supervised and unsupervised) for fault detection. For example, the authors can take a look at these articles:

https://doi.org/10.1080/00423114.2022.2103436

https://doi.org/10.1080/23248378.2022.2096132

2-     Where are the location of sensors 1,2 and 3 in figure 1?

3-     What is the difference between faults 1, and 2 in figure 3?

4-     What is the difference between conditions 1 and 2 in section 6.1?

5-     In line 490 what are setting 3 and 4?

6-     In section 6.5, it is not clear how the authors obtained crest factor, shape factor, kurtosis, …

Reviewer 2 Report

Comment 1: Abstract part of the redundant description too much, which can be simplified.

Comment 2: There are some problems with the image format and title layout, and some parts of the article layout are not suitable, please modify.

Comment 3: Please give a general explanation of the formula involved in the feature extraction method mentioned in the paper.

Comment 4: In the Experimental design, the specific reasons for faults 1, 2, 3 and 4 are not explained. Please briefly explain what the specific faults are.

Comment 5: The figure 43 mentioned on page 6 is nowhere to be found in the article.

Comment 6: The time domain features extracted by Pearson correlation analysis mentioned in the conclusion can reveal health and failure conditions. Please explain the specific working principle.

Comment 7: Please check the spelling and grammar of English words.

Round 2

Reviewer 1 Report

My remarks have been addressed properly. I think, it is OK now and can be published.

Reviewer 2 Report

All the comments have been considered in the revised manuscript, it can be accepted now.